# OpenReview forum: "Deep Bayesian Filter for Bayes-Faithful Data Assimilation"
_ICML.cc/2025/Conference — ICML 2025 poster_

### Official Review · Reviewer_gEdD · 2025-03-10

**Overall Recommendation:** 4

**Summary:**

In order to address the challenge of nonlinear filtering, the present work proposes _Deep Bayesian Filter (DBF)_, which leverages a learnable inverse observation operator (IOO) to transform the problem into a linear filtering problem, where one can apply standard Kalman methodologies. The cases of linear and nonlinear dynamical systems are treated differently: In the linear dynamics setting, having access to the IOO directly enables linear updates in physical space. In the nonlinear setting, an extra mapping to a high-dimensional latent space is required, where the hidden dynamics become approximately linear.  Training is performed once offline in a manner similar to VAEs. The model is tested on various benchmark dynamical systems, both linear and nonlinear, to demonstrate the robustness and performance gain achieved by DRF over classical filters.

## Update after rebuttal
The authors have adequately addressed my main concern regarding the scalability of the algorithm during the rebuttal period. Therefore, I have increased my evaluation of the paper from a 3 to a 4.

**Claims And Evidence:**

The claims in this paper are supported by clear and convincing evidence. In particular, they identify examples where the proposed model, DBF, significantly outperforms both classical DA methods and Dynamic VAEs. Results are presented clearly and demonstrate the benefits of using DBF.

**Essential References Not Discussed:**

The authors cover related literature in Section 2.6 fairly extensively. I would also point out that there is a line of works that has been coming out recently that use generative modelling techniques for data assimilation. For example,

- A Score-based Nonlinear Filter for Data Assimilation, Bao et al. (2023)
- DiffDA: A Diffusion Model for Weather-scale Data Assimilation, Huang et al. (2024)
- Score-based Data Assimilation, Rozet and Louppe (2024)

**Experimental Designs Or Analyses:**

The experimental setup is sound, evaluating the methods and baselines on commonly used benchmarks. The analysis of the experiments is also reasonable, and the arguments made are convincing. However, I would again point out the lack of comparison in the ability of the proposed model to perform UQ. In addition, some ablation study on the various components of DBF would be useful to have as well. For example, an ablation on the number of hidden state dimensions (in the case of nonlinear dynamics), a comparison of the two training strategies, etc... I would also think that having a discussion on what happens when you observe only a fraction of the components of Lorenz 96 model would be valuable too, as this is a common setting in DA (since it is rare in practice to observe the full state).

**Methods And Evaluation Criteria:**

The datasets used to evaluate the method are appropriate for the application at hand. The authors mainly evaluate the RMSE, which, however, can be limiting as the ability to quantify uncertainties is also crucial in data assimilation. Regarding uncertainty quantification, only the Jeffreys divergence of normalised errors with the standard Gaussian is presented in the example of the nonlinear pendulum. However, I believe that it is also worth evaluating other UQ metrics, such as the negative log-likelihood and the CRPS, which are commonly used metrics to compare the UQ ability of DA methods.

**Other Comments Or Suggestions:**

- I believe the $r_{true}(h_t | o_t)$ at the end of page 2 is supposed to be $r_{true}(z_t | o_t)$.
- I think the "rightmost table" at the end of page 5 is referring to Table 1, which is not "rightmost".
- It will be better to add standard deviations in the results in Table 3, as done in the other tables.

**Other Strengths And Weaknesses:**

__Strengths:__
- The presentation and organization of the paper are well thought-out, making it easy to follow.
- The experiments are fairly extensive, covering a range of examples where DBF can be beneficial over classical filters.
- The "Bayes-faithful" property of DRF is appealing, which is missing in most deep learning-based approaches. However, I also wonder how "faithful" it is in terms of its ability to approximate the true non-Gaussian posterior.

__Weaknesses:__

I think the biggest flaw of the proposed method is in its scalability to high-dimensional settings, where computation of the filtered mean and covariance (1)--(2) becomes intractable due to the cubic/quadratric scaling (with respect to dimension) in computational cost/memory. This is also the problem with Kalman filters in high dimensions, which is why ensemble methods like EnKF and ETKF were proposed in the first place. Especially when considering SSMs with nonlinear dynamics, DBF requires lifting the dynamics to an even higher dimensional space, where applying (4)--(5) will become too expensive to run, or otherwise one has to settle for latent dynamics in a lower dimensional space, where results may not be very accurate. Hence, the applicability of the method lies in low-to-moderate dimensional systems, in which case other classical Kalman-based methods like the Extended Kalman filter (EKF) and Unscented Kalman (UKF) filter can be applied. In particular, the UKF is known to perform well and robustly in such settings, hence, I believe it is only fair to compare with this, too.

**Questions For Authors:**

- Can the DBF framework deal with partial observations (i.e., only observing a few components of the L96) and observation operators that change with time (e.g., observing different components at each time step)? I believe these are important cases to consider as they occur commonly in practice. However, especially in the case where the observation operator changes with time, this does not seem straightforward, as one needs to amortize the inverse observation operator across various observation operators.
- In the setting of nonlinear dynamics, is there a general criteria for choosing the latent space dimension? i.e., how do we know a priori what latent space dimension should be chosen for the hidden dynamics to be approximately linear? It would be interesting to see an ablation study in the hidden dimension size.

**Relation To Broader Scientific Literature:**

The present work builds on a body of literature that aims to augment data assimilation/stochastic filtering with deep learning methodologies. However, in contrast to previous efforts like KalmanNet or methods based on dynamical VAEs, which use RNNs to model the latent dynamics, the present work uses explicit updates by assuming linearity of dynamics in the latent space. This helps prevent difficulties in training of RNN-based methods arising from problems such as vanishing gradient and Monte-Carlo approximation of the ELBO.

**Theoretical Claims:**

The main contribution of the paper is in the model proposal and empirical evidence of it working in certain settings where classical DA methods tend to struggle. There are no proofs to check.

---

> ### Author Rebuttal · Authors · 2025-04-01
>
> We deeply appreciate Reviewer gEdD for their thoughtful and constructive feedback. Below, we specifically address the reviewer’s important concerns.
>
> - Scalability in High-Dimensional Settings:
>
> We have conducted experiments varying latent dimensions in both the double pendulum and Lorenz96 settings, as reported in Appendix E of the original manuscript. Although we have not explicitly tested DBF in higher-dimensional physical systems, we expect that the latent dimensions required to accurately model the system dynamics would be substantially lower than the actual physical dimensions due to significant redundancy in degrees of freedom in large simulations.
> Moreover, even if higher dimensions were required, DBF remains resilient against explosive growth in computational demands. This aspect is detailed clearly in section C.1 of the Appendix, highlighting DBF’s efficient parameterization strategy that enables linear scaling with respect to latent dimension size, significantly alleviating the computational bottlenecks associated with traditional methods.
>
> - Partial Observations and Time-Varying Observation Operators:
>
> DBF finds no difficulty in handling partial observations. We need specific strategies to apply DBF for observation operators that vary over time. Amortizing a single inverse observation operator (IOO) across different observation operators can be indeed challenging. Future work could explore methods to learn conditional IOOs or multiple specialized IOOs to effectively handle scenarios with time-varying observation operators.
>
> - Criteria for Choosing Latent Space Dimension:
>
> The choice of latent space dimension primarily depends on the complexity of the underlying dynamics. In practice, we recommend selecting the smallest latent dimension that achieves good validation performance. Indeed, we have already conducted an extensive ablation study on latent dimension sizes in Appendix E of our manuscript. The results offer valuable insights and suggest practical heuristics for balancing model performance and computational efficiency.

---

> > ### Comment · Reviewer_gEdD · 2025-04-02
> >
> > I thank the authors for the clarifications.
> >
> > I understand that the authors have chosen a particular parameterization for the latent linear dynamics operator $A$ to make the prediction cost scalable in dimensions. However, the filtering step still involves taking matrix inversions (equations (4)-(5)) or determinants (should appear in the KL divergence computation in (6)), which is still going to be expensive. Could the authors please comment on this? The authors consider latent dimensions up to $O(10^3)$ in the appendix, which is still amenable to computation. However, if we require larger latent dimensions $> O(10^4)$ to adequately capture the dynamics (e.g. we are modelling a weather system), then how is computation managed?

---

> > > ### Author Response · Authors · 2025-04-03
> > >
> > > Thank you very much for the question.
> > >
> > > Although equations (4) and (5) involve the inversion of matrices such as $(A\Sigma_t A^T + Q)$, $G_{\theta}(o_t)$, and $V$, these are all block diagonal matrices given that $A$, $Q$, and $G_{\theta}(o_t)$ are block diagonal. Specifically, $A$ and $G_{\theta}(o_t)$ are maintained as block diagonal matrices, and the process noise covariance $Q$ is expressed as a diagonal matrix. Since the initial covariance matrix $\Sigma_{t=1}$ is block diagonal, it follows by induction (see equation (5)) that subsequent covariance matrices $\Sigma_{t}$ remain block diagonal. Consequently, rather than inverting one large matrix, we invert many small matrices — for example, in our implementation, we invert $10^4$ small $2\times 2$ matrices (i.e., when the full latent dimension is $2\times 10^4$). This structure significantly reduces the computational complexity of the filtering step.
> > >
> > > A similar argument applies to the computation of the KL divergence. Consider two Gaussian distributions:
> > >
> > > $q(h) = \mathcal{N}(\mu_1, \Sigma_1) \quad \text{and} \quad p(h) = \mathcal{N}(\mu_2, \Sigma_2),$
> > >
> > > where $\Sigma_1$ and $\Sigma_2$ are block diagonal, with each block being a $2 \times 2$ matrix, and there are $N$ such blocks. The KL divergence is given by:
> > >
> > > $\text{KL}[q\|p] = \frac{1}{2}\left[\log\frac{|\Sigma_2|}{|\Sigma_1|} - 2N + \operatorname{Tr}(\Sigma_2^{-1}\Sigma_1) + (\mu_2 - \mu_1)^T \Sigma_2^{-1}(\mu_2 - \mu_1)\right].$
> > >
> > > Due to the block diagonal structure, this expression can be factorized into a sum over the blocks:
> > >
> > > $\text{KL}[q\|p] = \sum_{i=1}^{N} \frac{1}{2}\left[\log\frac{|\Sigma_{2,i}|}{|\Sigma_{1,i}|} - 2 + \operatorname{Tr}(\Sigma_{2,i}^{-1}\Sigma_{1,i}) + (\mu_{2,i} - \mu_{1,i})^T \Sigma_{2,i}^{-1}(\mu_{2,i} - \mu_{1,i})\right].$
> > >
> > > This factorization means that the KL divergence computation is reduced to summing over many small, $2 \times 2$ matrix computations, making the process computationally manageable even for large latent dimensions.
> > >
> > > In summary, the block diagonal structure ensures that both the matrix inversion in the filtering step and the determinant computations for the KL divergence remain efficient, even if the latent dimension scales to $O(10^4)$ or beyond. In fact, this good scaling is another characteristic of our methodology, making it very promising for applications on very high dimensional problems, as required in data assimilation problems in natural sciences.

---

### Official Review · Reviewer_VtfJ · 2025-03-14

**Overall Recommendation:** 4

**Summary:**

This paper proposes a method for Bayesian filtering with nonlinear observations and dynamics, using a VAE specialized to Markov processes with linear latent dynamics. The result is a closed-form loss that can be optimized for both the encoder (latent state) and decoder (inverse observation operator, IOO).

**Claims And Evidence:**

The DBF shows strong improvement over relevant baselines.

**Essential References Not Discussed:**

None noted

**Experimental Designs Or Analyses:**

No concerns

**Methods And Evaluation Criteria:**

The experiments are a challenging set of nonlinear filtering tasks.

**Other Comments Or Suggestions:**

There are many settings where we don’t care about $z$ and just want to predict $o$. In that case the linear method of eq 6 can be used even if the mapping from $z$ to $o$ is nonlinear, because we only need $h$ while $z$ can be ignored.

Last line of page 2: I think you mean $h_t$ to be $z_t$ since $h$ has not been introduced yet.

Fig 1 caption: (c) should be (b)

**Other Strengths And Weaknesses:**

Despite its flexibility the IOO still assumes the log-likelihood is quadratic in $z_t$ which is a major simplification for many applications.

**Questions For Authors:**

I believe the Koopman operator works even for non-Markov processes: the sequence $h_t = g(z_t))$ can be made Markov even if $z_t$ is not (e.g., if $z$ is an AR-$k$ process for some $k>1$). Does this make DBF applicable in non-Markov settings?

Is the virtual prior $\rho$ necessary? Conceptually it’s strange because the prior is already present as $p(z_t|o_{1:t-1})$. Mathematically it’s redundant because it could be absorbed into $G$ as $(G_\theta(o_t)^{-1} - V^{-1})^{-1}$.

**Relation To Broader Scientific Literature:**

Relevant literature is discussed and it’s clear how the present method advances over previous ones.

**Theoretical Claims:**

I have convinced myself of the relevant derivations. The framework is very elegant.

In strategy 1, I think it would help to give more details on learning the map $z\mapsto A$ or $A\mapsto z$ from samples of $z_t$.

---

> ### Author Rebuttal · Authors · 2025-04-01
>
> We gratefully acknowledge Reviewer VtfJ for the positive evaluation and insightful feedback.
>
> - Direct prediction of $o_t$:
>
>  We agree and will explicitly clarify that Eq. (6) is particularly useful for cases where direct observation prediction is sufficient, even when the mapping from $z$ to $o$ is nonlinear.
>
> - Applicability to non-Markov settings:
>
>  We confirm that even if $z_t$ is not Markov, DBF extends naturally if there exist a Koopman embedding that satisfies $h_t = g(z_t))$, where $h_t$ is Markov. We will clarify this explicitly in our revision. We would like to thank VtfJ again for the insightful comment.
>
> - Virtual prior $\rho$:
>
>  We recognize that our explanation was insufficient. We have now clarified this point in the main text and provided additional details in Appendix A.4.
>
> We would like to thank VtfJ again for acknowledging the value of our methodology.

---

> > ### Comment · Reviewer_VtfJ · 2025-04-05
> >
> > Thanks for the replies. I think this will be a good paper.

---

### Official Review · Reviewer_zP6r · 2025-03-14

**Overall Recommendation:** 2

**Summary:**

The authors propose a novel variational method for data assimilation that constructs its variational family by replacing the non-linear observation model by a linear-Gaussian observation model whose mean and covariance are parametrized by a neural network.
If the dynamics of the prior are also linear-Gaussian, then the variational family is a linear-Gaussian state-space model in which is tractable using the Kalman filtering recursions. Inspired by Koopman operator theory, the authors extend the method to nonlinear dynamics by learning latent linear dynamics in a higher-dimensional space that map to the physical dynamics through a learned nonlinear transformation.
This transformation then makes the original method for linear dynamics with nonlinear observation models applicable.
The approach is shown to be competitive to classical DA methods (EnKF, ETKF) and multiple deep learning-based approaches on three benchmark problems covering both linear and nonlinear dynamics.

## Update after rebuttal
I am still very concerned with the introduction of the auxiliary prior in the derivation of the method, which I feel significantly impacts clarity (cf. review `VtfJ`). Given how pivotal this derivation is for the rest of the paper, I do not feel comfortable to raise my score without having seen the updated section of the paper.

**Claims And Evidence:**

- page 2, lines 68-69: The way this is written makes it sound like the Kalman filter only supports time-invariant dynamics and observation models, which is clearly not the case

**Essential References Not Discussed:**

None that I am aware of.

**Experimental Designs Or Analyses:**

While the experimental design seems sound, some of the choices made appear somewhat contrived. For instance, this includes the nonlinear observation operator $o_{t, j} = \min(z_{t, j}^4, 10) + \epsilon$, whose physical significance is unclear, and the choice of the moving MNIST problem as the main benchmark problem for linear dynamics. The object tracking experiment in the appendix seems to be a better choice, as it is much more interpretable.

**Methods And Evaluation Criteria:**

- page 6, lines 285-288, left: The definition of success seems arbitrary. What is the motivation for choosing exactly this threshold?

**Other Comments Or Suggestions:**

- page 1, lines 14-15, right: Does the term "test distribution" refer to the approximate filtering/smoothing posteriors provided by the respective methods?
- page 2, line 109, right: The use of $h_t$ in the context of linear dynamics is somewhat confusing, should this maybe read $r_\text{true}(z_t \mid o_t)$? This also applies to the first paragraph of Section 2.5.
- page 3, line 127, left: Missing "model" in "linear-Gaussian state-space *model* (LGSS*M*)"
- page 3, line 122, left: "Panel (b)" instead of "Panel (c)"
- page 3, line 128, right: Why is the physical state referred to as a "teacher signal" here? This term is not previously defined.

**Other Strengths And Weaknesses:**

Section 2.3 has significant clarity issues as it seems that the likelihood $p(o_t \mid z_t)$ is replaced by a posterior distribution $r(z_t \mid o_t)$ over the unobserved quantity. While this is remedied by dividing by a prior $\rho(z_t)$ "virtually introduced for the IOO", it is unclear what the significance of this prior is in the context of the dynamics model, which already induces a prior $p(z_t)$ over the state. The fact that the covariance matrix of $\rho(z_t)$ is fixed at $V = 10^8 I$ suggests that it is in fact irrelevant for the algorithm (see also equations 4 and 5) and only needed for the internal logic of the exposition.
I would hence like to suggest an equivalent exposition of the algorithm that avoids the virtual $\rho(z_t)$ prior and seems somewhat easier to follow: The authors aim at deriving a tractable variational family for a state-space model with linear-Gaussian transition model and nonlinear observation model. As noted in the paper, the nonlinear observation model hinders tractability of the original model and should hence be replaced, e.g. by a linear-Gaussian one in the variational family. Hence, inspired by the idea of an inverse observation operator, the variational family approximates the observation model by $p(o_t \mid z_t) \approx q(o_t \mid z_t) \propto \mathcal{N}(f_\theta(o_t); z_t, G_\theta(o_t)),$ which is linear-Gaussian. The dynamics of the variational family are simply given by the dynamics of the prior, i.e. $q(z_1) \coloneqq p(z_1)$ and $q(z_{t + 1} \mid z_t) \coloneqq p(z_{t + 1} \mid z_t)$. Then the conditional distribution $q(z_t \mid o_{1:t})$ can be computed by the Kalman filter recursion as in the paper. And used to compute the ELBO.
Note how this "likelihood-approximation" perspective renders the unintuitive virtual prior $\rho(z_t)$ unnecessary.
I'm very open to a discussion about resolving the clarity issues in this section and will likely raise my score should this be addressed satisfactorily.

**Questions For Authors:**

- Strategies 1 and 2 don't seem mutually exclusive. Which one should be used in which case? An experimental analysis comparing the two strategies might grant a lot of insight into the differences.
- Did you try to apply the approach to time-invariant dynamics? At least for linear dynamics or when using Strategy 2 for nonlinear dynamics, the method seems to be applicable to time-varying dynamics.
- What is meant by "marginalizing over $h_t$ with this emission model"? Surely this marginalization can't be performed in closed form?

**Relation To Broader Scientific Literature:**

The related work section is extensive and gives a detailed account of the commonalities and differences of the present work to the broader scientific literature.

**Theoretical Claims:**

- There are no theorems that require a proof.
- I did not check the derivations of the ELBOs.

- page 4, lines 177-180, left: It is unclear what is meant here. Is this an additional approximation introduced to make training tractable? Normally one is only at liberty to drop $z_{1:t}$ from $q(h_t \mid o_{1:t}, z_{1:t})$ in case $h_t$ and $z_{1:t}$ are conditionally independent given $o_{1:t}$, which does not seem to be the case here.

---

> ### Author Rebuttal · Authors · 2025-04-01
>
> We sincerely thank Reviewer zP6r for the thoughtful comments and constructive suggestions.
>
>
> - Clarification on Section 2.3
>
> We would like to respectfully clarify a key point regarding Section 2.3, which may have caused confusion due to insufficient clarity in our original exposition.
> The reviewer pointed out:
>
> “Section 2.3 has significant clarity issues as it seems that the likelihood p(ot∣zt)p(o_t \mid z_t) is replaced by a posterior distribution r(zt∣ot)r(z_t \mid o_t) over the unobserved quantity.”
>
> We acknowledge that the current presentation may have inadvertently led to this misunderstanding, and we appreciate the opportunity to clarify our intention.
> To be precise, our method does not replace the likelihood $p(o_t \mid z_t)$ with the posterior distribution $r(z_t \mid o_t)$. Rather, $r(z_t \mid o_t)$ is defined via Bayes' rule as being proportional to the product of the likelihood and a prior distribution $\rho(z_t)$:
> $r(z_t \mid o_t) \propto p(o_t \mid z_t) \cdot \rho(z_t).$
> No approximation is introduced in this definition itself. However, the challenge arises when we aim to perform recursive Bayesian inference, where we must evaluate:
> $p(z_t \mid o_{1:t}) \propto p(o_t \mid z_t) \cdot p(z_t \mid o_{1:t-1}).$
> Even if $p(o_t \mid z_t)$ is Gaussian, the dependence of its parameters on $z_t$ can make the posterior $p(z_t \mid o_{1:t})$ analytically intractable, particularly when the mean or covariance is a nonlinear function of $z_t$. To address this, we exploit the identity:
> $p(z_t \mid o_{1:t}) \propto \frac{r(z_t \mid o_t)}{\rho(z_t)} \cdot p(z_t \mid o_{1:t-1}),$
> and assume that $r(z_t \mid o_t)$, $\rho(z_t)$, and $p(z_t \mid o_{1:t-1})$ are all Gaussian distributions. This ensures that $p(z_t \mid o_{1:t})$ remains Gaussian and can be computed in closed form, thereby maintaining tractability of the filtering process.
> The approximations in our method thus lie in the following modeling choices:
>
> 1. We model $r(z_t \mid o_t)$ as a Gaussian distribution whose mean and covariance are parameterized by neural networks, denoted as $f_\theta(o_t)$ and $G_\theta(o_t)$, respectively.
>
> 2. The auxiliary prior $\rho(z_t)$ is assumed to be a Gaussian with fixed mean and fixed (large) covariance.
>
> In principle, both the parameters of $r(z_t \mid o_t)$ and those of $\rho(z_t)$ could be optimized during ELBO maximization. However, as shown in Equations (4) and (5), if the neural networks $f_\theta$ and $G_\theta$ are sufficiently expressive, fixing the parameters of $\rho(z_t)$ does not limit the representational capacity. The variability needed for inference is effectively captured by the learned functions $f_\theta(o_t)$ and $G_\theta(o_t)$. The large covariance of $\rho(z_t)$ ensures that its support covers the latent space broadly enough for effective approximation.
> We acknowledge that this important design rationale was not sufficiently explained in the original manuscript. We will revise Section 2.3 to clearly state the role of $\rho(z_t)$, clarify that it is not a replacement for the dynamics-induced prior $p(z_t)$, and explain why it can be treated as a fixed auxiliary distribution without loss of generality.
> We truly appreciate the reviewer’s insightful suggestion. We are confident that incorporating this clarification will significantly enhance the clarity and theoretical rigor of the manuscript.
>
> - Clarification regarding the approximation in $q(h_t \mid o_{1:t}, z_{1:t})$
>
> We appreciate the reviewer’s insightful question regarding the approximation introduced in the variational distribution.
> As correctly noted, in general, $q(h_t \mid o_{1:t}, z_{1:t}) \neq q(h_t \mid o_{1:t})$. In our method, we do introduce an approximation by replacing the former with the latter—i.e., we approximate the richer distribution $q(h_t \mid o_{1:t}, z_{1:t})$ using a simplified form $q(h_t \mid o_{1:t})$ for tractability.
> This choice restricts the expressiveness of the variational family, but it still provides a valid lower bound on the marginal log-likelihood, and therefore constitutes a legitimate ELBO formulation.
> The equality $q(h_t \mid o_{1:t}, z_{1:t}) = q(h_t \mid o_{1:t})$ would only strictly hold if the mapping from $z_t$ to $o_t$ were deterministic and invertible—i.e., when $z_t$ is fully determined by $o_t$. Since this is not the case in our setting, we acknowledge that the replacement is indeed an approximation.
> Nevertheless, this design allows for a computationally feasible training procedure, and we empirically find that it strikes a good balance between tractability and performance. We will clarify this point in the revised manuscript to avoid potential confusion.

---

> > ### Comment · Reviewer_zP6r · 2025-04-04
> >
> > ### Clarification on Section 2.3
> > > We would like to respectfully clarify a key point regarding Section 2.3, which may have caused confusion due to insufficient clarity in our original exposition.
> >
> > Thank you for the clarification, but I believe I already understood all of this from the paper. I have to admit that the formulation
> >
> > > [...] it seems that the likelihood p(ot∣zt)p(o_t \mid z_t) is replaced by a posterior distribution r(zt∣ot)r(z_t \mid o_t) over the unobserved quantity [...]
> >
> > was poorly chosen. However, I find the main point of my rebuttal unaddressed and hence would like to reiterate. My main concern is that the virtual prior makes the exposition very confusing (see also review VtfJ) and the method can be derived in a vastly simpler fashion without it.
> >
> > From equations (4) and (5) we can see that $\rho(z_t)$ has virtually no impact on $\mu_t$ and $\Sigma_t$, as $V^{-1} m = 0$ and $V^{-1} = 10^{-8} I \approx 0$. Your rebuttal suggests that this is by design, i.e., a maximally uninformative prior with a large support is sought. The derivation of the algorithm that I outlined in my review arrives at this result without the need for $\rho$. I will expand upon this here: The authors seek tractable variational families $q(z_t \mid o_{1:t})$ and $q(z_t \mid o_{1:t-1})$ for use in variational inference. These can be constructed by directly approximating the non-linear-Gaussian likelihood $p(o_t \mid z_t)$ with the linear-Gaussian likelihood $q(o_t \mid z_t) \propto \mathcal{N}(f_\theta(o_t); z_t, G_\theta(o_t))$ induced by the IOO (here the normalization is irrelevant, since the authors are interested in the conditionals of the variational family). Now we define
> > $$q(z_{1:T}, o_{1:T}) = q(o_1 \mid z_1) p(z_1) \prod_{t = 2}^T q(o_t \mid z_t) p(z_t \mid z_{t - 1}),$$
> > and, since $q(o_t \mid z_t)$ is linear-Gaussian, we can compute $q(z_t \mid o_{1:t})$ and $q(z_t \mid o_{1:t-1})$ with the Kalman filter. For instance, this yields $q(z_t \mid o_{1:t}) = \mathcal{N}(z_t; \mu_t, \Sigma_t)$ with
> > $$\mu_t = \Sigma_t (A \Sigma_{t - 1} A^\top + Q)^{-1} A \mu_{t - 1} + G_\theta(o_t)^{-1} f_\theta(o_t)$$
> > and
> > $$\Sigma_t^{-1} = (A \Sigma_{t - 1} A^\top + Q)^{-1} + G_\theta(o_t)^{-1}.$$
> > Since reviewer VtfJ also had doubts about the virtual prior, I would strongly encourage the authors to use this alternative derivation in the main paper, as it massively clarifies the exposition without any impact on the method.

---

> > > ### Author Response · Authors · 2025-04-07
> > >
> > > We sincerely thank the reviewer for the continued engagement and thoughtful follow-up. We are glad to hear that the clarification helped, and we appreciate your suggestion of an alternative derivation. We especially value your effort to propose a more concise formulation that may improve the clarity of exposition.
> > >
> > > However, we would like to respectfully point out a technical issue with the expression:
> > >
> > > $q(o_t \mid z_t) \propto \mathcal{N}(f_\theta(o_t); z_t, G_\theta(o_t)).$
> > >
> > > At first glance, this form may seem justified by the relationship:
> > >
> > > $p(o_t \mid z_t) \propto \frac{r(z_t \mid o_t)}{\rho(z_t)},$
> > >
> > > where $r(z_t \mid o_t) \propto p(o_t \mid z_t) \cdot \rho(z_t)$. Since $\rho(z_t)$ is independent of $o_t$, when interpreting $p(o_t \mid z_t)$ as a function of $o_t$, it is valid to write:
> > >
> > > $p(o_t \mid z_t) \propto r(z_t \mid o_t).$
> > >
> > > However, even if $r(z_t \mid o_t)$ is Gaussian in $z_t$, this does $\textbf{not}$ imply that $p(o_t \mid z_t)$ is Gaussian in $o_t$. More precisely, for any Gaussian emission model $p(o_t \mid z_t)$—whether linear or nonlinear—it must take the form:
> > >
> > > $p(o_t \mid z_t) = \mathcal{N}(o_t; \mu(z_t), \Sigma(z_t)) \propto \exp\left(-\frac{1}{2}(o_t - \mu(z_t))^\top \Sigma(z_t)^{-1}(o_t - \mu(z_t))\right),$
> > >
> > > which defines a density over $o_t$ with parameters depending on $z_t$. In contrast, the expression
> > >
> > > $q(o_t \mid z_t) \propto \mathcal{N}(f_\theta(o_t); z_t, G_\theta(o_t))$
> > >
> > > defines a density over $f_\theta(o_t)$, not over $o_t$, and is not generally interpretable as a valid probability density function over $o_t$ unless $f_\theta$ is invertible and appropriately constrained—which we do not assume.
> > >
> > > Therefore, while we appreciate your proposal and understand its motivation, we believe your formulation overlooks a key point: the tractability of our methodology does not arise from defining a Gaussian emission model over $o_t$, but from expressing $p(z_t \mid o_{1:t})$ as a Gaussian in $z_t$ via the expression:
> > >
> > > $p(z_t \mid o_{1:t}) \propto \frac{r(z_t \mid o_t)}{\rho(z_t)} \cdot p(z_t \mid o_{1:t-1})$
> > >
> > > with all terms on the right-hand side being Gaussians in $z_t$.
> > >
> > > In our approach, this is made possible by approximating $r(z_t \mid o_t)$, $\rho(z_t)$, and $p(z_t \mid o_{1:t-1})$ all as Gaussian distributions in $z_t$, which results in recursive update rules that are not only analytically tractable, but also generalize the classical Kalman filtering updates.
> > > More specifically, when the functions $f_\theta(o_t)$ and $G_\theta(o_t)$ are chosen appropriately, our update rules can recover the standard Kalman filter equations as a special case. Therefore, our formulation can be seen as a principled and flexible extension of Kalman filtering that enables inference in settings with nonlinear and learned observation models, while preserving closed-form computation at each time step.
> > >
> > > We acknowledge that the role of the auxiliary prior $\rho(z_t)$ may initially appear redundant, and we are grateful that Reviewer zP6r pointed out this source of confusion. We will revise the manuscript to clearly explain that $\rho(z_t)$ serves a functional purpose in the derivation—namely, enabling us to treat the ratio $r(z_t \mid o_t)/\rho(z_t)$ as a valid unnormalized likelihood over $z_t$—and why it cannot be skipped without undermining the theoretical grounding of the recursive inference.
> > > Once again, we thank the reviewer for raising this important point and for prompting us to improve the clarity of our exposition.

---

### Decision · Program_Chairs · 2025-05-01

**Decision:**

Accept (poster)

**Comment:**

This paper proposes a deep learning approach to Bayesian filtering in nonlinear discrete-time systems.
The introduction of a Gaussian inverse observation operator along with variational inference ensures the assimilation can take place in a linear space (thus Gaussian inference).

The paper is generally well received but some points should be addressed. Namely:

* The details regarding the likelihood approximation via inverse observation operator need to be significantly clarified. I.e. is the current formulation required to ensure that you know the approximation of p(z | o) up to and incuding normalization? or some other subtle point?
* The definition of "success" in the empirical validation ought to be more clearly motivated.
* The scalability of the method ought to be more carefully explained.